# Intersectionality and benefit receipt: The interplay between education, gender, age and migration background

Jos Slabbekoorn[1]*, Ineke Maas[1,2], J. Cok Vrooman[1,3]

1 Department of Sociology, Utrecht University, Utrecht, The Netherlands, 2 Department of Sociology, Vrije Universiteit, Amsterdam, The Netherlands, 3 The Netherlands Institute for Social Research | SCP, The Hague, The Netherlands

* j.slabbekoorn@uu.nl

**Data Availability Statement:** Data cannot be shared publicly because this paper makes use of microdata data from the Social Statistical Datasets (SSD). The data used in the analyses for this paper contain data on all individuals registered in the

## Abstract

This study examines differences in benefit receipt using an intersectional approach. Intersectionality theory emphasizes the importance of the interplay of multiple social dimensions. Taking this as a starting point, the paper investigates how different combinations of three demographic variables plus education buffer or amplify benefit receipt and thereby create relatively advantaged and disadvantaged groups. Administrative data were used, sourced from Dutch registers that provide accurate and detailed information on benefit receipt for the entire Dutch population, including small and hard-to-reach segments. Multilevel Analyses of Individual Heterogeneity and Discriminatory Accuracy (MAIHDA) are performed to assess which intersectional groups are relatively advantaged or disadvantaged with respect to benefit receipt. Intersectional group differences are more pronounced for social assistance than for unemployment insurance. Complex combinations of education, gender, age and migration background are required to better understand differences in benefit receipt, especially for unemployment insurance.

## 1. Introduction

As in most affluent welfare states, in the Netherlands benefit schemes have been created to reduce the consequences of unemployment among the labor force. These benefits (i.e., social assistance and unemployment insurance) aim to help individuals and households to meet their needs and sustain their standard of living to a certain degree [1]. In some social groups, the incidence of benefit receipt is elevated due to systemic inequities and discrimination mechanisms [2]. Various studies consistently find women, older people, people who followed vocational training and people with a migration background to be more dependent upon benefits [2–6]. Therefore, these social groups might be at greater risk to experience the negative consequences because–in spite of the monetary support provided by these schemes–benefit periods tend to increase the risk of poverty and to lower people's wellbeing [7, 8].

Prior research on benefit receipt has typically often relied on the implicit assumption that the advantages and disadvantages from various social dimensions are independent of one

Netherlands, and therefore can potentially identify individuals. Furthermore, these data are owned by a third-party organisation, which does not allow the sharing of these data. These data sets comprise information sources from 48+ Dutch government registers. Under certain conditions, these data are accessible for statistical and scientific research. For further information: microdata@cbs.nl.

**Funding:** This research was funded by a grant (2018-945: Project 'Intersectionality, Resources and Institutions: Determinants of Social Benefit Receipt'), awarded by Instituut Gak, an equity fund that subsidises projects related to social security and the labor market in the Netherlands, to J. Cok Vrooman. The funder did not have any role in study design, data collection and analysis, decision to publish, or preparation of the manuscript.

**Competing interests:** NO authors have competing interests.

another. There are a few notable exceptions, which considered non-additive accumulation of disadvantage by running separate models for men and women or migrants and natives [9, 10]. However, a growing body of intersectional literature argues that social dimensions constitute overlapping co-determinants of (dis)advantages [11]. From that perspective, previous studies may have under- or overestimated the (dis-)advantages faced by certain people, if they ignored the interdependence of multiple social characteristics, and an intersectional approach to studying benefit receipt seems warranted.

Intersectional approaches [12, 13] comprise a theoretical and analytical framework that considers multiple social dimensions simultaneously (e.g., gender, race, class and age). It is argued that the combination of such characteristics creates a unique social landscape that shapes the social identity and reality of individuals. The key idea behind intersectional approaches is that (dis-)advantages from multiple social dimensions do not simply add up but can influence each other. The multiple jeopardy hypothesis argues that being a member of multiple marginalized social groups can have larger negative effects than the simple combination of negative effects from all social group memberships [14, 15]. This amplification implies an added penalty for combining disadvantage from multiple marginalized group memberships, resulting in higher relative disadvantage and an increased risk of benefit receipt. Likewise, it is possible that membership of multiple marginalized social groups compensates part of the disadvantage that a specific social dimension might bring. For instance, women with a migration background seem to be able to partially buffer their [16], and it indicates that certain multiple groups can experience a relative advantage through specific combinations of risk factors.

Intersectional quantitative studies are scientifically relevant because they provide a more nuanced understanding of social inequality that accounts for the ways in which multiple forms of oppression intersect and interact. Recently, quantitative approaches in the analysis of intersectional inequalities are becoming increasingly popular. Thus far, these studies have mainly focused on labor market discrimination at the intersection of gender and migration background [16, 17], or people's health at the intersection of age, migration background and SES [18–20]. Intersectional insights may also be relevant for policy makers. Rather than focusing on broader population groups (e.g., young people, older people or people with a migration background) which is currently common practice, intersectional insights may help to focus efforts of social policy on specific subgroups (e.g. older people with a migration background) with an especially high incidence of benefit receipt.

The first contribution of this paper is theoretical. It adds to the literature by developing a typology of six combinations of disadvantages and advantages, which identify patterns of social inequality that are often overlooked in traditional quantitative approaches. By using an intersectional and mechanism-based approach, we theorize how different forms of disadvantage and advantage might combine to shape social outcomes. Hereby, we move beyond simple descriptions of patterns of inequality and towards a more sophisticated understanding of the underlying mechanisms that drive these patterns. Secondly, we introduce a new and efficient way of analyzing the many interactions implied by intersectional research to mainstream sociological research. Multilevel Analyses of Individual Heterogeneity and Discriminatory Accuracy (MAIHDA) has–with very few exceptions–only been used in medical studies [21]. As we will show it is a parsimonious method to estimate the advantages and disadvantages of the 163 intersectional groups that we distinguish. Third, we use register data covering the complete population of the Netherlands, that include small and hard to reach population segments (e.g., older low-educated women with a migration background), that may be especially disadvantaged. We will therefore also provide descriptive results that can guide policy interventions. By identifying the specific ways in which different groups experience inequality, policymakers

can more effectively target interventions to address the root causes of social inequality and promote greater equity.

Therefore, this article explores the complexities of inequality in social benefit receipt by answering two questions: (1) *to what extent do intersectional differences (i.e., unique combinations of education, gender, age and migration background) occur in benefit receipt in the Netherlands*? And (2) *which intersections are relatively disadvantaged or advantaged (i.e., show non-additive negative or positive effects on benefit receipt)*?

In this article, register data from the System of Social Statistical Datasets (SSD) were used, which are made available by Statistics Netherlands (CBS). This database consists of interlinked datasets on the entire population of the Netherlands. These data contain reliable and detailed information on benefit receipt. Multilevel Analyses of Individual Heterogeneity and Discriminatory Accuracy [MAIHDA, 18] are used to analyze the intersectional nature of benefit receipt. This article focuses on two programs that provide financial support to people who have experienced a partial or complete loss of employment: benefits based on the Unemployment Insurance Act (Werkloosheidswet, or WW) and general Social Assistance (Bijstand). Dutch unemployment insurance is available to individuals who involuntarily lost paid employment. Its duration is limited and in order to be eligible, people must have worked 26 weeks out of the previous 39; if they are entitled, the benefit lasts between 3 and 36 months, depending on their employment history. Unemployment insurance benefits levels are 70–75% of a person's previous earnings from waged employment (up to a maximum amount), and entitlements are not means-tested. Social assistance provides financial support to adults in households whose combined income is below the statutory social minimum and whose assets do not exceed 7,575 euro for single person households, and 15,150 euro for household with more than one adult person. This includes housing property of which the first 63,900 euros of equity on the house is exempted. In line with the European Union directive for non-discriminatory social policies, the entry requirements for social assistance and unemployment insurance are not dependent upon nationality and these benefits are accessible to all (previously employed) individuals legally residing in the Netherlands. The only exceptions pertain to immigrants with a short-term residence permit who recently came to the Netherlands. Together, these benefit schemes serve as an integrated social welfare system that covers the risk of unemployment due to economic factors. Individuals can transition from unemployment insurance to social assistance when they have surpassed the maximum period of unemployment insurance and meet the eligibility criteria for social assistance. Since the determinants for receipt of these benefits may be vastly different, they are analyzed separately in our main analyses.

## 1.1 Theoretical background: Social inequalities in benefit receipt

In the subsequent section, we will first present theoretical arguments on disparities in benefit receipt related to education, gender, migration background, and age separately. This gives an overview of the current state of social welfare research, and explicates which social groups have a generally higher incidence of benefit receipt. Following this, we will present an intersectional theoretical argument, offering an overview of six pathways through which (dis-)advantages stemming from multiple social identities may intersect and subsequently influence the relative advantage or disadvantage with respect to the likelihood of benefit receipt for intersectional groups.

**1.1.1 Education.** A substantial sociological and economic literature analyses the effect of educational attainment on employment [22]. While the knowledge gained during education can be considered a valuable resource in itself [23] educational systems also form an important

locus for the (re-)production of social class [24, 25]. Educational qualifications serve not only as an indicator of knowledge, skills and competences, but also as an indicator for the composite value of embodied cultural capital. This implies a poorer competitive position on the labor market for lower educated individuals, who consequently are less likely to find secure and stable employment. Social segregation further strengthens those dynamics. Individuals establish long-lasting relationships and accumulate social capital at school. Later in life on the labor market, jobseekers can receive job referrals, support, and recommendations from their contacts which improve their employment chances [26, 27]. Lower educated individuals may have fewer valuable contacts who could help them find a secure and stable job. Furthermore, lower educated people have less healthy lifestyles, since they experience poorer life and work circumstances (i.e., living in unhealthy neighborhoods and physically demanding jobs), have more limited opportunities for healthy behavior [28]. This puts them at higher risk to be without a job due to health reasons. Based on the reasoning above, it is expected that lower educated people are generally more likely to receive a benefit than higher educated people.

**1.1.2 Gender.** Although the gap in educational attainment between men and women has recently closed, women consistently hold lower status and less secure jobs than men [29]. Living up to gendered expectations of society at large and their own social circles, women might make different career and study choices, as they anticipate motherhood-related changes in employment later in life [30–33]. Many women seek employment in sectors that are motherhood-friendly [34]. In addition, women may be more willing to take-up lower status and less secure jobs in order to combine family and work commitments [e.g., through part-time or zero-hour contracts: 29]. Some women are voluntarily jobless and live in a traditional household where the man is the sole breadwinner. In the case of divorce or a deceased partner, these women in particular are prone to be on social assistance, as their inactivity on the labor market makes them ineligible for other benefits. Additionally, this labor market status and childcare responsibilities can reduce their opportunities to find stable employment. Lastly, women face discrimination on the labor market contingent on the risk of work absence during pregnancy and periods of paternal leave [35, 36], combined with gendered expectations of employers [17]. In turn, this can make it more difficult to find stable and secure employment [37]. All in all, women face several disadvantages that increase their likelihood of unemployment and the risk of benefit receipt. Therefore, it is expected that women are, in general, more likely to receive a benefit than men.

**1.1.3 Age.** At the end of their professional careers, people have accumulated (work-specific) experience, knowledge, skills, contacts, and status. However, some knowledge and skills of people in their 50s and 60s may have become obsolete or less valuable on the labor market [38]. For instance, some older people may experience greater difficulty in acquiring skills that are essential in the increasingly digitalizing labor market [39]. Additionally, older people may face age discrimination, which affects employers' hiring decisions [40]. This may, in particular, reduce their chances of re-entering the labor market after losing their jobs [41, 42], thus increasing the likelihood for long-term unemployment. Lastly, since health tends to deteriorate with age [43], older people are more likely to lose employment due to health reasons. All in all, it is expected that older people are generally more likely to receive a benefit than less senior people.

Young people are at the start of their professional careers, and often are still in the process of acquiring (work-specific) skills, knowledge, and contacts. Therefore, they typically hold unstable, short-term, lower status jobs, such as traineeships [44]. Therefore, the risk for young people to become unemployed is greater; but due to their limited work experience, they are less likely to be eligible for Dutch unemployment insurance benefits. If they are entitled, young people can rely on unemployment insurance benefit for a shorter period of time, since its

duration is based on the individual employment history. Additionally, young people who enter into benefit receipt are more likely to obtain help from regional employment offices. Active Labor Market Policies in the Netherlands, as in other EU countries, specifically target younger age groups, as they are deemed easier to help to work [45]. Therefore, young people can get out of benefit receipt more easily. In sum, and despite their position in the labor market, it is expected that younger people are generally less likely to receive a benefit than mid-aged and older people.

**1.1.4 Migration background.** There are considerable differences between migrant groups and natives, which may translate into diverging incidence rates of benefit receipt [46, 47]. On the one hand, poorer language proficiency of some groups of first generation migrants may on average lead to a lower take- up of benefits. However, findings regarding the take-up of benefits are mixed; where some studies find comparable take-up of benefits between migrants and natives [48, 49], others find lower take-up of benefits [50]. Recent migrants may also have a too short employment history to be eligible for unemployment benefits, but with the exception of those who are in the Netherlands shorter than 3 months, all legal migrants are eligible for social assistance. We do not expect that eligibility and differential take-up will lead to a substantial lower incidence of benefit receipt for migrants compared to natives.

On the other hand, migrants are more likely to have deficiencies in resources, leading to poorer labor market outcomes [51]. There is a substantial gap in Dutch language proficiency between several migrant groups and natives, although this is less among second generation migrants [52]. Since proficiency in the Dutch language is required for many jobs, the employment chances for people with a migration background are likely to be reduced, and consequently it might be expected that they have a higher likelihood of benefit receipt. Studies also consistently find that–even when the levels of resources are equal–people with a migration background face additional forms of disadvantage on the labor market [53, 54]. Research shows that they are less likely to be invited for job interviews, probably because the prevailing ethnic stereotypes inform the choices that employers make [55]. It is found that ethnic discrimination particularly affects individuals with a migration background from countries that are less socio-economically developed than the Netherlands, such as Turkey and Morocco. On these grounds, it is expected that people with a migration background are more likely to receive a benefit than people without a migration background. In addition, we expect the incidence of benefit receipt to depend on country of origin, and to be highest among first generation migrants, and lower among second generation migrants.

## 1.2 Intersectionality and benefit receipt

Intersectionality theory argues that memberships of multiple social groups (e.g., gender, migration background, education, age) are interconnected and should be understood in relationship to each other [12]. Thus, the meaning and implications of being a member of a certain social group (e.g., being a migrant) is conditional on other social dimensions (e.g., people's age, education, or gender). From an overarching perspective, the intersectional literature provides a framework through which the combination of disadvantages and advantages can be studied [15]. The core idea behind this strand of literature is that the combination of advantages and disadvantages stemming from various social dimensions can be more complex than a simple sum of parts. In academic fields like gender studies, an intersectional lens is commonly used to study multiply disadvantaged groups [56]. However, theoretically, there are six possible non-additive ways in which disadvantages and advantages from two social identities can be interconnected (numbered A–F in Fig 1). The remainder of our theoretical argument will further elaborate upon the ways disadvantages and advantages could combine and

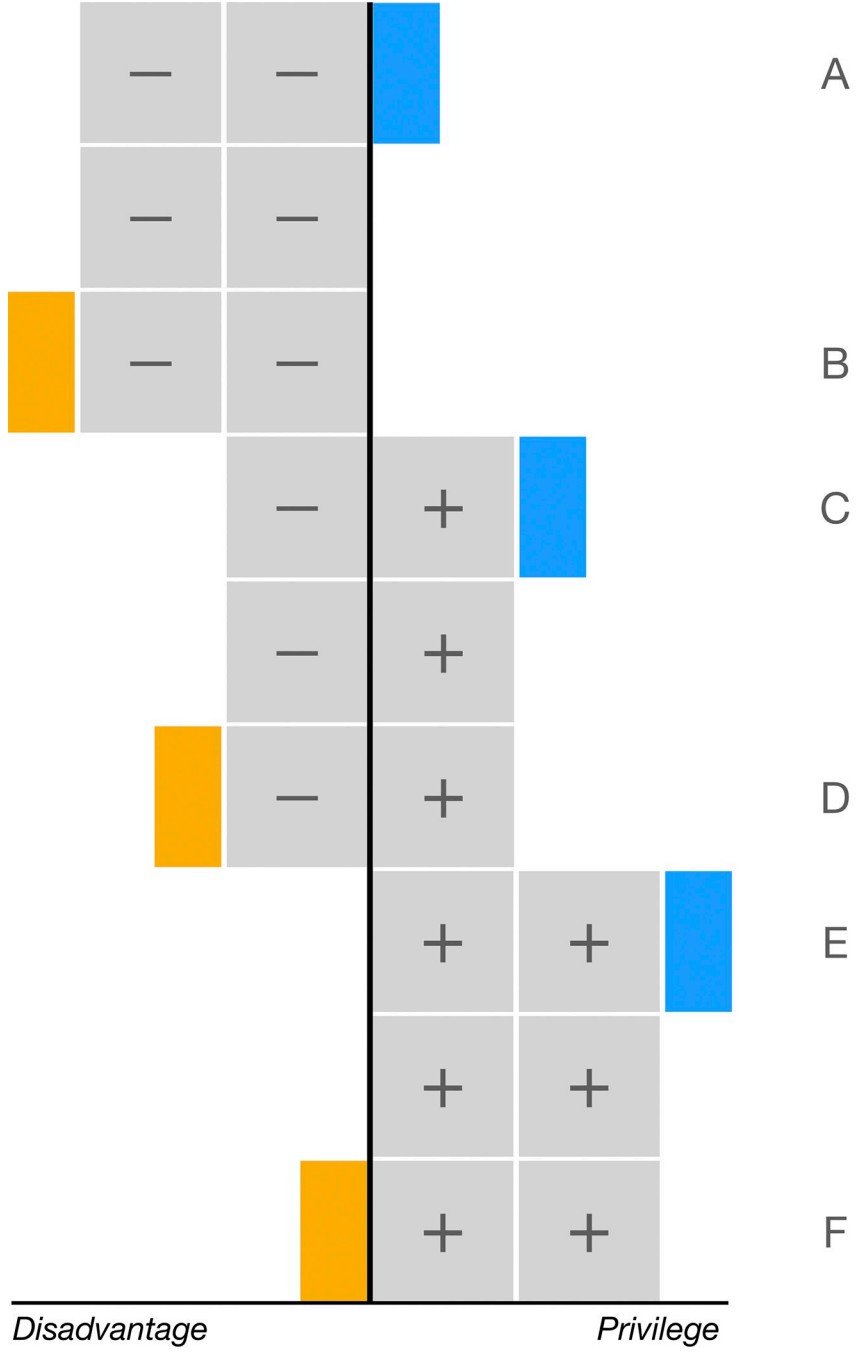

**Fig 1. Examples of intersectional combinations of advantages and disadvantages. Note:** Gray squares with a [−] denote disadvantageous social characteristics. Gray squares with a [+] denote advantageous social characteristics. Blue rectangles denote the extra rewards for relatively advantaged groups. Orange rectangles denote the extra penalties for relatively disadvantaged groups. In addition to the six interconnected combinations of advantages and disadvantages (number A through F), the additive combinations (i.e., those that do not comprise extra rewards or penalties) are included as a baseline.

mutually influence each other. First, three cases where people are relatively disadvantaged (shaded orange in Fig 1) will be discussed. At these intersections, people are worse off than would be expected based on a simple combination of advantages and/or disadvantages [57]. In

essence, people face an extra penalty for their multiple group membership. Second, a discussion of three cases that could lead to a relative advantage (shaded blue in Fig 1) follows. These cases are better off than expected based on the simple combination of advantages and/or disadvantages.

Inherently, relative (dis)advantages stem from comparisons between intersections. This implies that a relative disadvantage for one intersection might be resulting from a relative advantage of another intersection. For example, the multiple jeopardy for black women in the US [56] can only arise in a social context where another group holds privileged positions–in this case white men. This double-edged sword principal always applies when comparing dichotomous cases, such as those that are discussed in the remainder of our theoretical argument. However, in higher dimensional intersectional comparisons–as is the case in our analysis–one relative disadvantage is not necessarily opposed to a specific advantaged group but could be caused by factors that particularly burden a specific intersectional group.

**1.2.1 Relative disadvantage.** King's [14] famous multiple jeopardy hypothesis comes to mind first when thinking of relatively disadvantaged groups. The multiple jeopardy hypothesis posits that simultaneously being a member of more than one marginalized group exceeds the negative effects of a simple addition of disadvantages. Disadvantages from multiple sources could amplify each other and result in an additional penalty that increases the risk of negative experiences (case B in Fig 1). The multiple jeopardy hypothesis [14, 58] was developed to study the marginalized position of black women in the United States. However, other intersectional groups might face a multiple jeopardy in similar ways, such as: lower educated migrants. Lower educated migrants are for example less fluent in Dutch than higher educated migrants. This could further aggravate the ethnic disadvantage on the labor market, which could make them more vulnerable for benefit receipt.

Furthermore, intersectional groups that combine an advantage and disadvantage could still be relatively disadvantaged (case D in Fig 1). These intersectional groups might not completely compensate their disadvantage with their advantage, since they might not be able to fully benefit from their advantaged position. A first example of such a case can be found in the classic work of [59], who show that black men benefitted less from having a higher educational attainment than their white counterparts. Due to factors like discrimination, black higher educated men were less likely to be employed and if they had found employment, they did so in lower status jobs than white men with the same education. A second example concerns lower educated men who benefit from gender privilege but are disadvantaged due to their educational attainment. For men, having a lower education might pose a relatively bigger disadvantage, since jobs that are typically held by lower educated men are more susceptible to being crowded out, than jobs typically held by lower educated women [60, 61]. Additionally, lower educated men work in more physically demanding jobs than lower educated women, which increases their risk of benefit receipt [62].

In a similar vein, multiply advantaged groups might not be fully able to benefit from their advantaged position (case F in Fig 1). Where multiple privileges can help to acquire positions of power and higher income, they might not help to provide additional protection against benefit receipt. And especially regarding benefit receipt, there might be diminishing returns to advantages. Once a social group is sufficiently advantaged to reduce the risk of benefit receipt (e.g., by being likely to hold a secure job), advantages from another social identity might only marginally lower the risk of benefit receipt. For example, the risk of benefit receipt of higher educated individuals is probably affected less by privilege from their gender- or native background-identity than the risk of lower educated individuals.

**1.2.2 Relative advantage.** Quantitative research on labor market discrimination that studied the intersection of gender and migration background found a reverse gender gap [[17];

case A in Fig 1]. Although women with a migration background combine two disadvantaged social characteristics, these studies show that they have a relative advantage compared to men with a migration background on the labor market. It is argued that discrimination based on ethnic stereotypes particularly impacts men, since their profile is more in line with the prevailing ethnic stereotypical prototypes [e.g., higher likelihood of having a criminal past, aggression, female unfriendly, see 63, 64]. Since women with a migration background fit these ethnic stereotypes less well, their ethnic penalty might be partially buffered [65]. Therefore, women with a migration background might be less vulnerable for benefit receipt compared to men with a migration background.

Similarly, intersectional groups that combine an advantage and disadvantage, might be able to overcome some of the disadvantage they experience. As such, these groups do better than expected based on the simple combination of their advantage and disadvantage (case C in Fig 1). For instance, higher educated older men might be able to turn their age disadvantage into an advantage. Older men may have profited from Matthew effects and have accumulated career success [66]. This career success consolidates in having increased job security [i.e., having a tenured job position, see: 67]. In turn, their age effect is more positive than for women; older women less often hold tenured job positions, since many women reduced working hours after childbirth.

Lastly, having multiple advantaged identities could be considered an advantage in itself (case E in Fig 1). And just as disadvantages may strengthen each other and thereby create an additional punishment, two advantages can be mutually reinforcing and bolster one's position even further [12, 15, 68]. White men in particular may experience mutually reinforcing advantages on the labor market. Since powerful people who make hiring decisions are (and traditionally have been) male with a native background, the labor market is an environment that may favor white men in two ways. First, employers may make homophily-based choices and prefer to hire candidates that are most similar to them (white men prefer to hire white men). Second, employers may prefer to hire candidates that are most similar to people that have formerly held similar positions. Since, a large share of high-status jobs is held by men, this makes it disproportionately likely that they will be hired in high status jobs [15, 69], in turn lowering their risk for benefit receipt.

In the section above, six examples of complex combinations of advantaged and disadvantaged social characteristics that could lead to a relative advantage or a relative disadvantage for certain intersectional groups were discussed. These examples mainly comprised intersections of two social characteristics and illustrated how these can be mutually influential. The focus on dyadic intersection was made intentionally to provide a simple overview of the general theoretical mechanisms that could illustrate the potential impact of different intersections on social outcomes. Additionally, not all possible combinations of age, gender, migration background and educational attainment were discussed because there is currently little theoretical work available for the intersections of concern in this article. A detailed discussion of all possible combinations of age, gender, migration background and educational attainment would forgo the exploratory aims of this article, where the aim of this article is to numerically identify relatively disadvantaged and advantaged groups.

## 2. Materials and methods

### 2.1 Data

For this article, register data from the Social Statistical Datasets (SSD) were used (Under certain conditions, these data are accessible for statistical and scientific research. For further information: microdata@cbs.nl). This database comprises over forty standardized and

interlinked administrative datasets, sourced from various Dutch registers [70]. In this paper we sourced data from PERSOONSTAB, SECMBUS and HOOGSTEOPLTAB which contain information that is provided by Dutch Municipalities (Person Population Register, Administration of Employee Insurance Schemes), and Governmental Administrations (Administrations of Employee Insurance Schemes, and the Social Security Bank) as well as individual tax and education data. These data provide a unique opportunity for quantitative intersectional analyses of benefit receipt because it contains detailed and reliable data on benefit receipt for all registered inhabitants of the Netherlands, including harder to reach populations.

The observation period was limited to start at 01-01-2006, and end at 31-12-2019 because data for some benefits were only included in the SSD as of 2006, and to exclude potential COVID19-related biases. Our sample was limited to employable individuals of working age using the following criteria: (a) they should be at least 25 years of age in 2019, (b) they should be younger than 60 years of age in 2006, (c) they should not be a student or pensioner (since these groups may not be eligible due to other social provisions they receive), and (d) they should at least have 3 years of observations. This resulted in a sample of approximately 6.5 million employable individuals of working age. Then, a stratified sample per intersectional group was drawn to facilitate the estimation of our analytical models. For privacy reasons, 17 intersectional groups were excluded, since they consisted of fewer than 100 individuals. From strata with fewer than 600 individuals half of the individuals were selected and from strata comprising 600 or more individuals 300 individuals were. This resulted in at an analytical sample of 41,599 individuals. This sampling procedure was repeated twice (not including the same individuals) to create a comparable training-sample, which was used for model calibration purposes.

## 2.2 Operationalization

**2.2.1 Dependent variables.** Two dependent variables were constructed: *social assistance benefit receipt* and *unemployment insurance benefit receipt*, using information from the national income register. This register contains information about individuals' major source of income per month. Lastly, this information was aggregated such that the variables indicate whether an individual had received a benefit (i.e., social assistance or unemployment insurance benefit) as a main source of income for at least three consecutive months during the observation period. Unlike many econometric studies on benefit receipt that analyze entry and exit rates [2–6], our operationalization focuses on the incidence of benefit use. This captures the net effect of entry and exit rates, reflecting who ultimately utilizes these programs. While acknowledging that social assistance typically targets households, we analyze individual data of social assistance receipt because all household members have to be eligible for social assistance in order to receive it, and we assume the household members will share the benefit.

**2.2.2 Independent variables.** *Gender* was operationalized as the registered gender in the 2019 register. Information from the 2019 register was used to include all potential changes in registered gender. In 2014 restrictions were lifted that prohibited changes in registered gender in the Netherlands. Therefore, the most recent record in the person registration is more likely to reflect the gender expression of an individual.

*Age groups* the following three age groups were constructed: young [25–34], middle [35–49] and old [50–59]. Observations of individuals were censored if they were younger or older than the outer range of their age group. For example, observations between 2006 and 2011 were excluded for individuals who turned 25 in 2012.

*Education* was operationalized to indicate whether an individual holds a university degree. Information from HOOGSTOPLTAB was used, which is a microdata source from Statistics

Netherlands based on administrative and survey data. Per individual, the first available record in the education register after they turned 25 years old was used. Since this data source suffers from systemic incompleteness in the registration of non-academic levels of education due to the gradual roll-out of the registration, and therefore, no distinction was made between other levels of educational attainment. People with an academic degree have the lowest incidence of benefit receipt [71]. In the past three decades the registration of education was implemented in the following chronological order: universities (1983), universities for applied science (in Dutch HBO, 1986), high schools (2003/04), schools for vocational training (in Dutch MBO, 2004/05). Some missing information was imputed by Statistics Netherlands using information from the Labour Force Survey.

*Migration background* was constructed using the registered country of birth of individuals and their parents from the person register. Eight origin groups were constructed: Dutch, Dutch Antillean, Moroccan, Surinamese, Turkish, Central and Eastern European, Other European and a miscellaneous group. These origin groups were chosen to provide more detailed information for the largest migration groups in the Netherlands, while retaining information on all people with a migration background from other countries. For all non-Dutch origin groups, it was distinguished whether an individual was a first-generation immigrant (foreign-born) or second-generation migrant (native-born with at least one foreign-born parent). For first-generation migrants, the individuals' country of birth was used to infer migration background. For second generation migrants, the birth country of the mother was used, except when the mother was born in the Netherlands, then the country of birth of the father was used to infer migration background. Individuals who were born in the Netherlands and whose parents were both born in the Netherlands were classified as Dutch natives. For the descriptive statistics of the variables used in our analyses, see Table 1.

## 2.3 Analytical strategy

In this article Multilevel Analyses of Individual Heterogeneity and Discriminatory Accuracy (MAIHDA) [18] were used, for which Bayesian logistic multilevel regression models were estimated in Stata 16.1 (for a more detailed and technical description of our analytical procedure see S1 File). MAIHDA has rapidly become the state-of-the-art method for intersectional quantitative research, as it overcomes some of the key challenges that quantitative intersectional analyses pose [72]. (a) Traditional models risk inflating false positives, especially for small groups. Our approach employs "shrinkage" to pull interaction effect estimates towards the mean, reducing this risk and ensuring reliable findings. (b) Explicitly modeling every interaction between all social dimensions can be cumbersome. Our approach treats "multiplicative" effects as random intercept variations, leading to a simpler and more efficient model. (c) Theoretically, situating individuals within intersectional social strata, as MAIHDA does, reflects the focus on group level processes of inequality, which aligns well with the existing intersectional and social stratification literature.

In our models, individuals were nested in intersectional strata. These intersectional strata comprise all unique combination of gender, age groups, migration background groups (incl. first and second generation) and educational attainment (respectively: 2×3×15×2). In our sampling procedure, 17 intersectional strata that had fewer than 100 members in the employable population were excluded, which resulted in 163 intersectional strata. Social assistance and unemployment insurance were modelled separately.

First, baseline models (indexed: 0) were estimated. These models only include the intercept (denoting the overall average incidence), as well as random intercepts (denoting the differences between the predicted incidence per intersectional stratum and the overall average

**Table 1. Descriptive statistics of the employable population and analytical sample.**

|  | Core-Workforce (%) | Analytical Sample (%)[a] |
|---|---|---|
| **Gender** | 49.99 | 50.21 |
| Female |  |  |
| Male | 50.01 | 49.79 |
| **Age** | 46.09 | 38.74 |
| Young |  |  |
| Middle | 35.50 | 35.70 |
| Old | 18.41 | 25.55 |
| **Migration Background** | 73.25 | 7.96 |
| Dutch |  |  |
| Dutch Antilles 1st gen. | 1.01 | 6.33 |
| Dutch Antilles 2nd gen. | 0.35 | 5.15 |
| Moroccan 1st gen. | 1.66 | 6.47 |
| Moroccan 2nd gen. | 0.87 | 4.19 |
| Surinamese 1st gen. | 1.69 | 7.40 |
| Surinamese 2nd gen. | 1.11 | 6.50 |
| Turkish 1st gen. | 1.88 | 6.85 |
| Turkish 2nd gen. | 1.05 | 4.36 |
| Eastern European 1st gen. | 1.66 | 7.20 |
| Eastern European 2nd gen. | 0.16 | 5.73 |
| Other European 1st gen. | 2.38 | 7.96 |
| Other European 2nd gen. | 2.21 | 7.96 |
| Other 1st gen. | 7.12 | 7.96 |
| Other 2nd gen. | 3.62 | 7.96 |
| **Education** | 12.51 | 44.16 |
| Academic |  |  |
| Non-Academic | 87.49 | 55.84 |
| **Benefit Receipt** | 9.95 | 16.83 |
| Social Assistance |  |  |
| Unemployment Insurance | 19.22 | 30.06 |

**Note:** For categorical or binary variables, the mean reflects the proportion. Due to rounding, the proportions of some categorical variables may not add up to exactly 100%. N(Analytical Sample) = 45,119, N(Core-Workforce) = 6,555,549. **Source:** Authors' own calculation based on non-public individual level register data from the Social Statistical Datasets (SSD) of Statistics Netherlands (CBS). **a:** The proportional size of Dutch natives is smaller compared to other origin groups in the analytical sample because for them the distinction between generations cannot be made.

incidence). These models were used to estimate the incidence of benefit receipt for all intersectional strata. The baseline models were also used to calculate the intraclass correlation (ICC). The ICC serves as a measure for discriminatory accuracy and tells how much of the variation in benefit receipt can be attributed to differences between intersectional groups.

Second, so called partially adjusted models were estimated, one for each social dimension that was used to construct the intersectional strata (indexed: 1–4). In these models, dummy variables for each social group per dimension were included. These models were used to calculate how much of the between strata variance can be explained by the respective social dimension, for which the proportional change of variance (PCVs, see S2 File).

Third, fully adjusted models were estimated, in which all additive effects of all social dimensions were included simultaneously (indexed: 5). These models were used to calculated how

much of the between strata variation in benefit receipt can be explained. The remaining stratum level variation in benefit receipt can be attributed to the complex combinations of social dimensions, which could lead to relative (dis-)advantages. The estimated random intercepts of this model denote the difference between the actual incidence per stratum and the predicted incidence based on the additive effects per stratum. In other words, the random intercepts of the fully adjusted model capture the relative (dis-)advantage per intersectional stratum. These are often referred to as the intersectional effect or multiplicative effect.

A possible further step in modelling intersectional differences in benefit receipt would be to add control, mediating or moderating variables. However, we refrain from doing so, because we first want to establish to what extent intersectional effects exist. In light of recent discussion of MAIHDA models regarding the interpretation of model estimates, we would like to repeat Evans and colleagues' [72] counterarguments. First, the fixed effect predictions are weighted for sample size and provide conservative estimates of grand means per social dimension. This is one of the advantages of MAIHDA–as it reduces the influence of the majority group in the sample in the estimates of the fixed effect–, which prevents that the majority group is used as a default with which other strata are compared. Second, the critique against interpretation of the fixed effect predictions in MAIHDA [73] focuses on specific simulated conditions, where group level variation is unrealistically high. In conditions with minor to moderate group level variation–like in the case of this article–grand means fall within the credibility intervals of the fixed effect predictions.

We ran additional analyses to assess the relation between the length of the observation likelihood and benefit receipt incidence (see S3 File). We do not find an association between duration and social assistance incidence. We do find a weak positive association with the incidence of unemployment insurance, however, this association disappears when we take age group into account. This suggests that the initial association with unemployment insurance was driven by age related difference in the duration of the observation period.

## 3. Results

### 3.1 Explained variances

Table 2 shows the results for Model 5, including the calculated ICCs and PCVs for social assistance and unemployment insurance benefit receipt (see S2 File for the "PCVs" of the partially adjusted models 1–4). For social assistance benefit receipt, it is found that group differences can explain 11.3% of the variation in incidence (ICCs = 0.113). This shows that there are substantial differences in social assistance receipt between intersectional groups; however, social group membership is by no means the sole contributing factor. The differences between intersectional groups can be largely explained in additive terms. 79.3% of the differences between intersectional groups are due to differences by gender, migration background, age and education PCVs = 0.793. The remaining 20.7% of intersectional group differences can be attributed to the complex combinations of social dimension associated (dis-)advantages. Men have a 1.6% lower social assistance benefit receipt incidence of social assistance than women. People aged between 35 and 49 have a 1.3% higher incidence than young people. People with an academic degree have a 14.3% lower incidence than people without an academic degree. The remaining 20.7% of intersectional group differences can be attributed to the complex combinations of social dimension associated (dis-)advantages. Compared to the overall incidence of social assistance receipt in or sample of 12.1%, these differences are moderate to substantial. Compared to the overall incidence of social assistance receipt in or sample of 12.1%, these differences are moderate to substantial.

**Table 2. Summary of additive effects for social assistance benefit receipt and unemployment insurance benefit receipt.**

| | Social Assistance | Unemployment Insurance |
|---|---|---|
| **Intercept** | 0.312 | 0.414 |
| | (0.286; 0.342) | (0.404; 0.427) |
| **Gender** (ref. Female) | -0.016 | 0.024 |
| Male | | |
| | (-0.022; -0.009) | (0.015; 0.031) |
| **Migration Background** (ref. Dutch) | -0.075 | -0.071 |
| Dutch Antillean 1st Gen. | | |
| | (-0.087; -0.062) | (-0.089; -0.053) |
| Dutch Antillean 2nd Gen. | -0.125 | -0.096 |
| | (-0.132; -0.118) | (-0.117; -0.072) |
| Moroccan 1st Gen. | 0.085 | -0.024 |
| | (0.044; 0.115) | (-0.045; -0.002) |
| Moroccan 2nd Gen. | -0.095 | -0.102 |
| | (-0.105; -0.084) | (-0.115; -0.088) |
| Surinamese 1st Gen. | -0.011 | 0.006 |
| | (-0.029; 0.017) | (-0.012; 0.022) |
| Surinamese 2nd Gen. | -0.077 | -0.045 |
| | (-0.096; -0.057) | (-0.055; -0.037) |
| Turkish 1st Gen. | 0.001 | -0.032 |
| | (-0.026; 0.029) | (-0.047; -0.012) |
| Turkish 2nd Gen. | -0.142 | -0.083 |
| | (-0.153; -0.132) | (-0.095; -0.072) |
| Eastern European 1st Gen. | -0.084 | 0.046 |
| | (-0.103; -0.068) | (0.026; 0.064) |
| Eastern European, 2nd Gen. | -0.112 | -0.116 |
| | (-0.129; -0.089) | (-0.131; -0.100) |
| Other European 1st Gen. | -0.083 | 0.012 |
| | (-0.094; -0.070) | (-0.011; 0.033) |
| Other European 2nd Gen. | -0.124 | -0.061 |
| | (-0.148; -0.101) | (-0.074; -0.044) |
| Other 1st Gen. | 0.081 | 0.002 |
| | (0.037; 0.123) | (-0.011; 0.012) |
| Other 2nd Gen. | -0.105 | -0.076 |
| | (-0.125; -0.082) | (-0.087; -0.066) |
| **Age** (ref. Young) | 0.013 | 0.089 |
| Middle | | |
| | (0.003; 0.024) | (0.076; 0.105) |
| Old | 0.003 | 0.048 |
| | (-0.007; 0.014) | (0.034; 0.064) |
| **Education** (ref. Non-Academic) | -0.143 | -0.104 |
| Academic | | |
| | (-0.158; -0.131) | (-0.114; -0.093) |
| Log Likelihood | -17478.227 | -26582.606 |
| | (-17497.368; -17460.117) | (-26602.325; -26565.723) |
| ICC | 0.113 | 0.041 |
| | (0.090; 0.140) | (0.032; 0.052) |
| PCV | 0.793 | 0.517 |

(*Continued*)

**Table 2.** (Continued)

| | Social Assistance | Unemployment Insurance |
|---|---|---|
| | (0.743; 0.838) | (0.317; 0.670) |

**Note:** Averages of the fixed effects (as average marginal effects) posterior distributions, 95% credibility interval between parentheses. N(individuals) = 45,119. **Source:** Authors' own calculation based on non-public individual level register data from the Social Statistical Database (SSD) of Statistics Netherlands (CBS).

For unemployment insurance benefit receipt, 4.1% of the variation in incidence can be explained by intersectional group differences. Compared to social assistance a smaller fraction can be attributed to differences between intersectional strata and other contributing factors play a more important role in unemployment benefit receipt. Conversely, although differences between intersectional groups in terms of unemployment insurance benefit incidence are smaller compared to social assistance, the complex combinations of gender, migration background, age and education play a relatively larger role. A considerable fraction of the intersectional group differences (100–51.7 = 48.3%) is attributable to non-additive effects. Men have a 2.4% higher incidence of unemployment insurance benefit. Young people have an 8.9% lower incidence compared to people of middle age and a 4.8% lower incidence compared to older people. Older people have a 4.1% lower incidence compared to people of middle-aged. Academically educated people have a 10.4% lower incidence compared to people without an academic degree. These are moderate to substantial differences compared to the baseline incidence of unemployment benefit recipiency of 29.2%.

### 3.2 Predicted incidences

Fig 2 shows the predicted incidence of social assistance and unemployment insurance benefit receipt for all strata (numerical values are available in S2 File). Distinct patterns in social assistance utilization across demographic groups become apparent. Individuals with a migration background display a higher incidence of receiving social assistance compared to native strata. There are two exceptions: immigrants of European origin and second-generation immigrants with academic qualifications. They exhibit incidence rates similar to native strata. For social assistance benefit receipt, the lowest incidence is found among Dutch middle-aged women with an academic degree ($\hat{P}$ = 0.014; 1.4%). People with a Dutch or European background with an academic degree generally have the lowest incidence of social assistance. The highest incidence of social assistance benefit receipt is found among older women with a first-generation Moroccan migration background who did not have an academic degree ($\hat{P}$ = 0.595; 59.5%). Among the groups with the highest incidence of social assistance an over-representation of people with a migration background that have not had an academic education is found. Compared to the overall incidence of social assistance receipt in or sample of 12.1%, these differences are moderate to substantial.

Fig 2 shows that there is less variation in unemployment insurance incidence compared to social assistance incidence. The lowest incidence of unemployment insurance benefit receipt is found among older aged women with a first-generation Moroccan migration background who did not have an academic degree ($\hat{P}$ = 0.121; 12.1%). This group, as well as middle aged women with a Moroccan origin, show considerably lower incidences than other lower educated women with a first generation migration background. Among intersectional groups with the lowest incidence of unemployment insurance benefit young people with an academic degree are overrepresented. The highest incidence is found among older aged men with a first generation easter European migration background who did not have an academic degree ($\hat{P}$ =

## Social Assistance Incidence

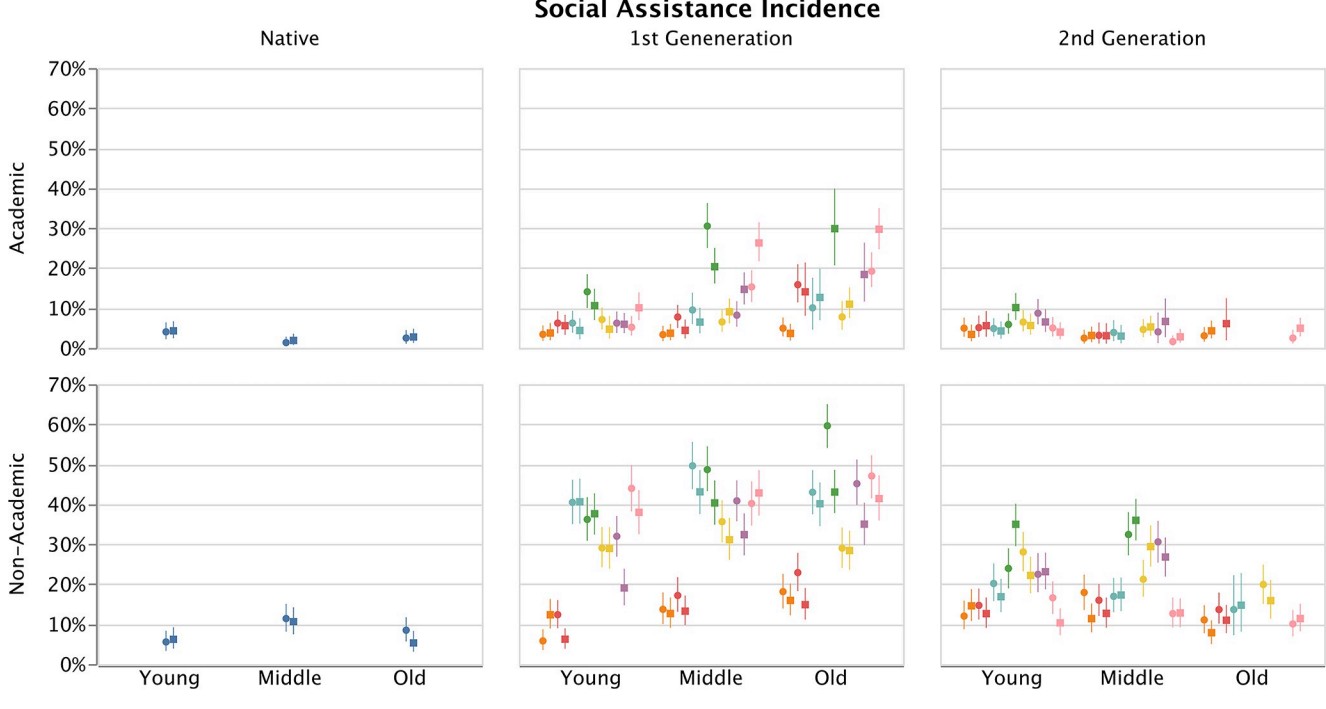

## Unemployment Insurance Incidence

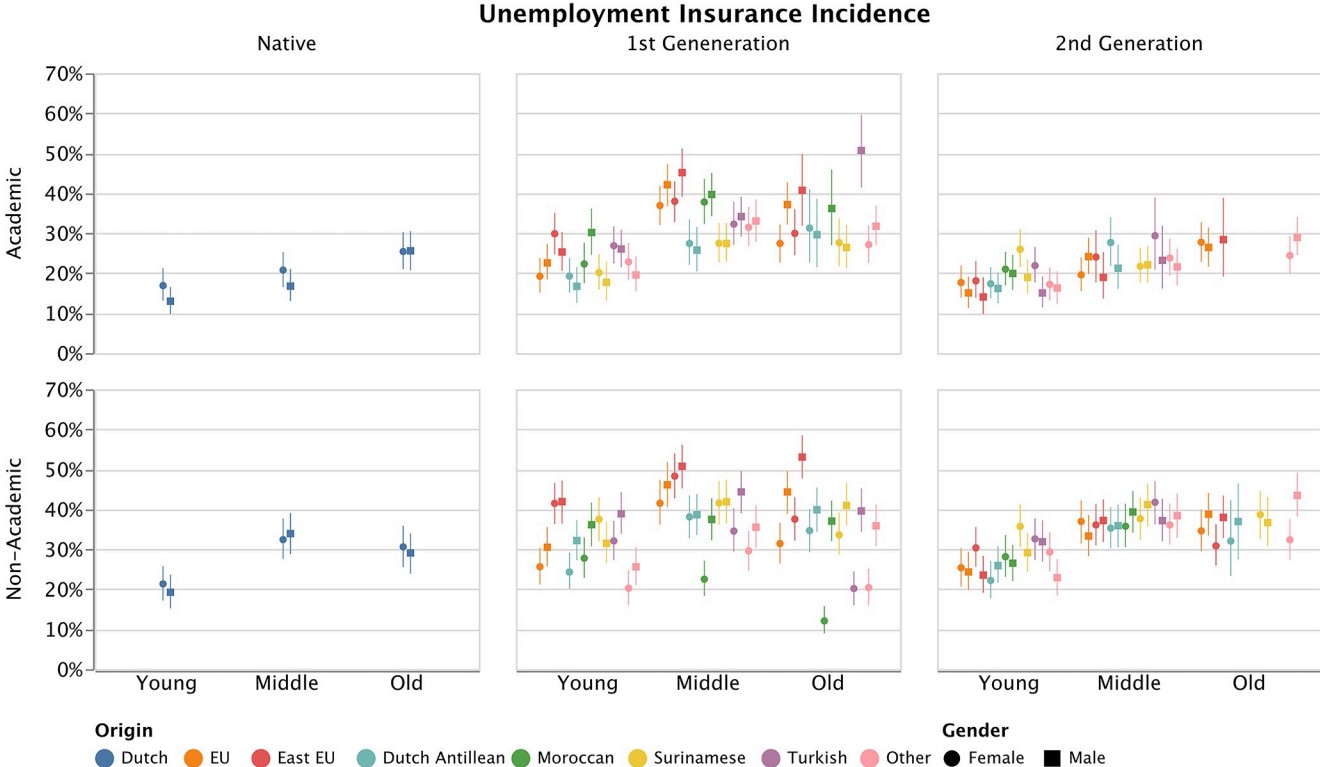

**Fig 2. Estimated incidence per stratum of social assistance benefit and unemployment insurance benefit receipt. Note:** Estimated incidence per stratum of baseline models (Model 0) based on burn-in depth of 10,000 iterations and posterior distributions 10,000 iterations. 95% credibility interval are displayed as error bars. Strata smaller than 100 individuals were omitted from the analyses, see the analytical strategy for a more elaborate discussion. N(individuals) = 45,119. **Source:** Authors' own calculation based on non-public individual level register data from the Social Statistical Database (SSD) of Statistics Netherlands (CBS).

0.530; 53%). Among the groups with the highest incidence of unemployment insurance, lower educated men with a European migration background are overrepresented. Additionally, the groups with an academic degree that had a high incidence of unemployment insurance all had an Eastern European background. These are moderate to substantial differences compared to the baseline incidence of unemployment benefit recipiency (29.2%).

### 3.3 Intersectional effects

The intersectional effects (i.e., the random intercept per intersectional stratum) are visualized in Fig 3 (for a numerical summary of the results see S2 File). The intersectional effect for most strata is not statistically different from 0. This means that the incidence of benefit receipt (of social assistance and unemployment insurance) for most strata is adequately explained using additive effects. However, for a considerable number of groups a significant intersectional effect is found. In these strata, the incidence of benefit receipt is lower or higher than would be predicted under additive assumptions. Thus, at these intersections social dimensions might be mutually influential co-determinants of benefit receipt, leading to a relatively higher or lower incidence of benefit receipt.

For social assistance, 65 out of 164 strata had an intersectional effect that was statistically distinct from 0, of which 27 had a negative intersectional effect (i.e., were relatively advantaged) and 32 had a positive intersectional effect (i.e., were relatively disadvantage), see Fig 3 and Table 3. For a complete overview see S2 File. Among the relatively advantaged groups, the following groups are overrepresented: academically educated Dutch individuals, academically educated women with a migration background and young people with a migration background. These groups have a lower incidence of social assistance than was expected using the

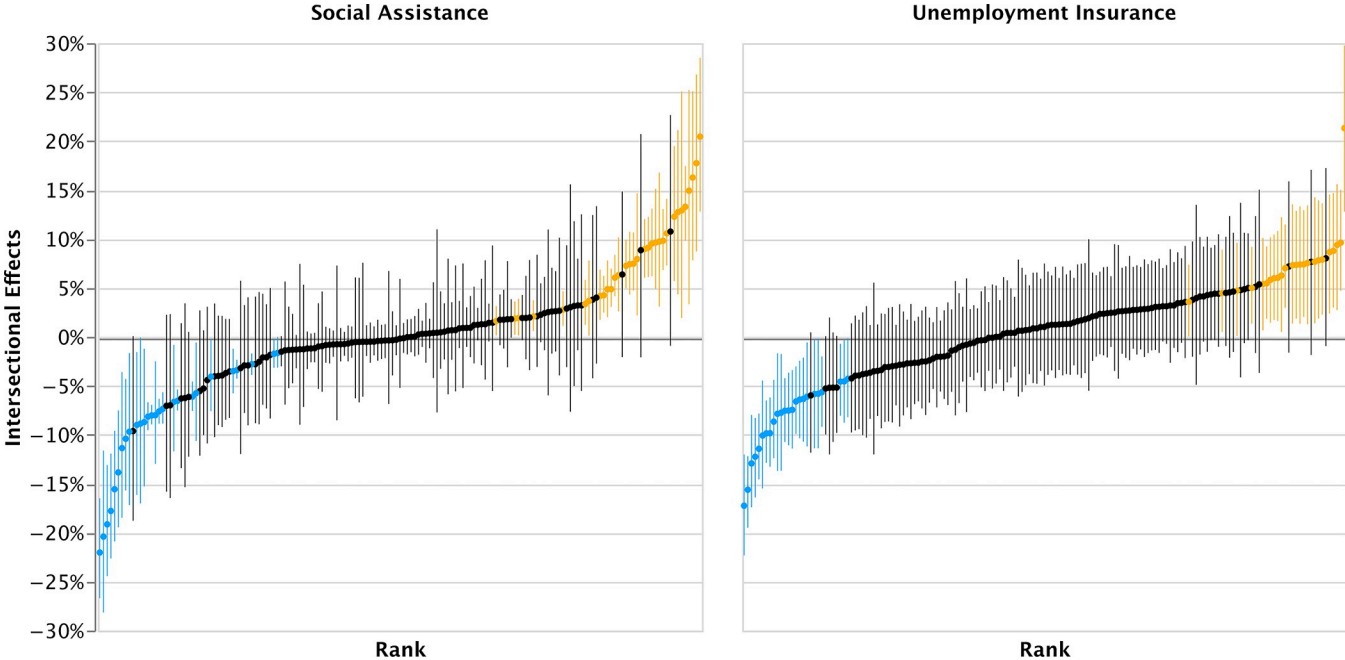

**Fig 3. Estimated intersectional effects (as average marginal effects) of social assistance benefit and unemployment insurance benefit receipt. Note:** (Orange = relative disadvantaged), (Blue = relative advantaged), and (Black = Not relatively (dis-) advantaged). Estimated posterior distributions of the random intercept terms of fully adjusted models (Model 5) based on burn-in depth of 10,000 iterations and posterior distributions 10,000 iterations. 95% credibility interval are displayed as error bars. The strata are ranked by the predicted intersectional effect for each respective benefit. N(individuals) = 45,119. **Source:** Authors' own calculation based on non-public individual level register data from the Social Statistical Database (SSD) of Statistics Netherlands (CBS).

**Table 3. Intersectional effects (as average marginal effects) for social assistance benefit receipt and unemployment insurance benefit receipt.**

| Background | Sex | Age | Education | AME | 95%CI | | Prop.[a] |
|---|---|---|---|---|---|---|---|
| | | | | Social Assistance | | | |
| Dutch | Female | Old | Academic | -0.191 | (-0.244; -0.131) | | 0.368 |
| Dutch | Male | Middle | Academic | -0.178 | (-0.226; -0.119) | | 1.897 |
| Dutch | Male | Old | Academic | -0.156 | (-0.209; -0.096) | | 0.752 |
| Dutch | Female | Young | Academic | -0.138 | (-0.195; -0.075) | | 2.331 |
| Turkish 1st Gen. | Female | Young | Academic | -0.114 | (-0.185; -0.036) | | 0.013 |
| Dutch | Male | Young | Academic | -0.104 | (-0.157; -0.043) | | 1.807 |
| Surinamese 1st Gen. | Female | Middle | Academic | -0.097 | (-0.172; -0.017) | | 0.022 |
| Turkish 1st Gen. | Male | Young | Academic | -0.090 | (-0.161; -0.016) | | 0.015 |
| Turkish 1st Gen. | Female | Middle | Academic | -0.089 | (-0.170; -0.001) | | 0.009 |
| Surinamese 1st Gen. | Male | Young | Academic | -0.087 | (-0.153; -0.012) | | 0.007 |
| Dutch Antillean 1st Gen. | Male | Young | Non-Academic | 0.098 | (0.068; 0.130) | | 0.200 |
| Dutch Antillean 1st Gen. | Male | Middle | Non-Academic | 0.106 | (0.073; 0.141) | | 0.160 |
| Turkish 2nd Gen. | Female | Young | Academic | 0.123 | (0.057; 0.195) | | 0.024 |
| Moroccan 1st Gen. | Female | Middle | Academic | 0.127 | (0.044; 0.211) | | 0.008 |
| Eastern European 1st Gen. | Male | Old | Academic | 0.129 | (0.019; 0.251) | | 0.003 |
| Dutch Antillean 1st Gen. | Female | Middle | Non-Academic | 0.133 | (0.098; 0.174) | | 0.159 |
| Moroccan 1st Gen. | Male | Old | Academic | 0.149 | (0.033; 0.252) | | 0.003 |
| Other 1st Gen. | Male | Middle | Academic | 0.163 | (0.078; 0.251) | | 0.115 |
| Eastern European 1st Gen. | Female | Old | Academic | 0.177 | (0.087; 0.268) | | 0.007 |
| Other 1st Gen. | Male | Old | Academic | 0.204 | (0.128; 0.285) | | 0.058 |
| | | | | Unemployment Insurance | | | |
| Dutch | Male | Middle | Academic | -0.172 | (-0.223; -0.120) | | 1.897 |
| Moroccan 1st Gen. | Female | Old | Non-Academic | -0.156 | (-0.195; -0.122) | | 0.116 |
| Dutch | Male | Young | Academic | -0.129 | (-0.174; -0.080) | | 1.807 |
| Moroccan 1st Gen. | Female | Middle | Non-Academic | -0.122 | (-0.164; -0.083) | | 0.376 |
| Other 1st Gen. | Female | Old | Non-Academic | -0.114 | (-0.145; -0.078) | | 0.536 |
| Dutch | Female | Middle | Academic | -0.101 | (-0.155; -0.045) | | 1.570 |
| Dutch | Male | Young | Non-Academic | -0.099 | (-0.129; -0.065) | | 13.688 |
| Turkish 1st Gen. | Female | Old | Non-Academic | -0.098 | (-0.133; -0.062) | | 0.130 |
| Other 1st Gen. | Female | Middle | Non-Academic | -0.087 | (-0.124; -0.044) | | 1.171 |
| Surinamese 1st Gen. | Male | Young | Academic | -0.079 | (-0.137; -0.017) | | 0.007 |
| Surinamese 2nd Gen. | Female | Young | Academic | 0.074 | (0.019; 0.129) | | 0.037 |
| Eastern European 1st Gen. | Male | Middle | Academic | 0.075 | (0.013; 0.134) | | 0.008 |
| Moroccan 1st Gen. | Male | Young | Academic | 0.077 | (0.014; 0.142) | | 0.008 |
| Eastern European 1st Gen. | Male | Old | Academic | 0.078 | (0.019; 0.139) | | 0.018 |
| Other 2nd Gen. | Male | Old | Academic | 0.079 | (0.023; 0.136) | | 0.041 |
| Eastern European 1st Gen. | Male | Middle | Academic | 0.087 | (0.024; 0.146) | | 0.059 |
| Moroccan 1st Gen. | Male | Middle | Academic | 0.088 | (0.030; 0.147) | | 0.017 |
| Moroccan 1st Gen. | Female | Middle | Academic | 0.094 | (0.027; 0.156) | | 0.008 |
| Other 2nd Gen. | Male | Old | Non-Academic | 0.096 | (0.047; 0.150) | | 0.251 |
| Turkish 1st Gen. | Male | Old | Academic | 0.213 | (0.128; 0.298) | | 0.003 |

**Note:** Averages of the intersectional effects (as average marginal effects) posterior distributions, 95% credibility interval between parentheses. N(individuals) = 45,119. In this table, only the intersectional effects are presented for the top and bottom 10 strata that were substantially different from zero. **Source:** Authors' own calculation based on non-public individual level register data from the Social Statistical Database (SSD) of Statistics Netherlands (CBS). **a:** This column gives the proportional size of the respective stratum in the full Dutch employable population.

additive model. It appears that all academically educated Dutch, irrespective of gender and age, are more likely to be able to avoid social assistance than could be predicted using the characteristics of their strata. The same is true for young higher educated first generation migrants from 'traditional' countries of origin (Suriname and Turkey). It seems that they benefit more from their educational attainment than other groups. Conversely, when looking at the relatively disadvantaged groups, generally older academically educated people with a migration background and younger lower educate people with a migration background were found. This seems to suggest that older migrant groups have benefited less from education-related advantage.

For unemployment insurance, fewer intersectional effects were found. Namely, 48 out of 164 strata had a statistically distinct intersectional effect, of which 24 were relatively advantaged and 24 relatively disadvantaged. Among relatively advantaged groups, again academically educated Dutch people were found. Also, groups of non-academically educated women with a migration background less often receive unemployment benefits than expected. These results suggest that Dutch individuals with an academic degree might experience an amplification of their advantage that yields an extra relative advantage. Migrant women might be able to buffer some of the ethnic penalty they experience, resulting in a relatively advantaged position compared to migrant men. In correspondence with the results for social assistance, mainly academically educated men with a first-generation migration background were found among the relatively disadvantaged groups for unemployment insurance. Ethnic discrimination might explain why they are less able to fully benefit from their academic qualifications.

## 4. Conclusion

This study examined group differences in benefit receipt in the Netherlands and identified notable disparities at the intersection of gender, age, education and migration background. The variation among intersectional groups in receiving social assistance benefits is more pronounced than for unemployment insurance benefits. However, the differences in social assistance benefit receipt could be largely explained by the combined additive effects of migration background and education level. In contrast, the variation in unemployment insurance incidence was more intricate, and required complex combinations of gender, migration background, age, and education to explain group-level variation more accurately. We find the highest incidence of social assistance among older women with a first-generation Moroccan migration background, and the lowest among Dutch middle-aged women with an academic degree. For unemployment insurance the highest incidence is found among older aged men with a first generation Eastern European migration background who did not have an academic degree and the lowest among older aged women with a first-generation Moroccan migration background who did not have an academic degree. The main aim of this paper was to develop a theoretical schema of six different types of intersectional effects and to describe the extent of intersectional inequalities in benefit receipt. We discuss our findings against the light of the theoretically distinguished types of intersectional advantage and disadvantage.

Intersectional differences are most often associated with the idea that membership of more than one marginalized group leads to disadvantages that exceed those expected when simply summing up the disadvantage of each group. We find very little evidence for this multiple jeopardy hypothesis [14, 15]. The intersectional groups that show most relative disadvantage almost all have academic education. This means that they are not generally disadvantaged, but they are less advantaged than one would expect based on their high level of education. All these groups have a migration background and most of them are men. These findings are consistent with findings from the classic study by Blau and Duncan [59] and suggest that migrants

may not fully benefit from academic qualifications, illustrating cases of diminished education related advantage for these subpopulations. They showcase our second type of intersectional effects in which a group that is disadvantaged in one respect cannot fully benefit from an advantaged position on another dimension. We also find some indications for the existence of our third type of intersectional relative disadvantage: the case that multiply advantaged groups are not fully able to benefit from all their privileges. First generation male immigrants from EU countries with academic education (often called expats) are more likely to receive unemployment benefits than one would expect based on their characteristics.

We also distinguished three types of intersectional relative advantage. According to the reverse gender gap hypothesis women with a migration background are less disadvantaged than to be expected based on these two disadvantageous characteristics [16, 17]. In line with this hypothesis, we find that strata of women with a migration background are often found in the relatively advantaged groups with regards to having a low likelihood of receiving unemployment insurance benefits. This can be explained by women with a migration background being able to mitigate some of their ethnic disadvantage. They experience less discrimination on the labor market and are less likely to lose their job, in line with findings from previous research [16, 17]. However, it is also possible that at these intersections women are more likely to take on the role of stay-at-home mother. As a result, a larger proportion of women in these groups may not qualify for unemployment insurance benefits due to their limited work experience. Our finding that young native Dutch men with academic education are even more likely to avoid social assistance and unemployment benefits than expected based on their favorable characteristics, supports our expectation that multiple advantages can strengthen each other. We also find this for middle aged and older men, and for women of all age groups. These are examples of the last type of intersectional effects in which advantageous characteristics (being native Dutch and higher educated) compensate for the (smaller) disadvantages of being older and female.

In conclusion, our findings suggest that previous studies on benefit receipt, that assumed that disadvantage or advantage accumulates in an additive manner, have underestimated or overestimated benefit receipt for certain intersectional groups. Our results demonstrate that age, gender, migration background, and education are interconnected factors that co-determine benefit receipt and should be considered together. However, intersectionality is not a simple cumulation of advantages or disadvantages, but more often comprises that groups that are disadvantaged in one respect are not able to fully benefit from some advantageous characteristic.

We did not study how intersectional effects in benefit receipt can be explained, but the most probable explanations are lack of resources, and discrimination. Higher educated first generation migrants may, for example, lack social capital needed to find a (new) job. But it is also possible that they have no difficulties in identifying interesting vacancies, but that they are discriminated when applying for jobs. Research has shown that both types of explanations are needed to explain additive effects, but it is still largely unclear how to explain intersectional effects. Future research is also needed to explore entry and exit dynamics of benefit receipt to better understand the intersectional inequalities underlying benefit dependency. Some groups might have more difficulty accessing benefits, others might experience a harder time to find re-employment and have a longer duration of benefit receipt.

Although the use of register data is one of the qualities of this study, it comes with several drawbacks regarding the operationalization of certain independent variables. First, due to systemic missingness resulting from the gradual roll-out of the education register across all levels of education, the operationalization of educational attainment was limited. Ideally, we would have distinguished within the group without tertiary education, since groups that have a

secondary education might be especially vulnerable to benefit receipt. In this light, the intersectional effects might be slightly overestimated for intersectional groups with an over-representation of people without a tertiary education. Second, the sociodemographic information contained in the person register might not perfectly reflect the self-identification of individuals. Individuals might not identify with their registered gender because their gender identity might not fit within the male/female dichotomy, or they might not have had the opportunity to change their registered gender. In a similar vein, individuals might not identify themselves with the country of origin (of their parents). Qualitative methods to study intersectional inequalities in benefit receipt are more suited to study such anti-categorical self-identification of individuals. Additionally, other aspects that concern migration, such as reasons for leaving the country of origin and the duration of residence after migration could provide valuable insights as to which migrant groups have a higher incidence of benefit receipt.

Furthermore, it is imperative to contextualize the results presented in this paper in light of one notable qualification. Due to data limitations, this study cannot take into account individuals' eligibility for benefit programs. Our assumption now is that people who do not receive benefits do not have a need for it because they have a paid job or other resources to get by. However, some individuals meeting the criteria for either social assistance or unemployment insurance might not have received a benefit for various reasons (i.e., not applying, not obtaining, and not qualifying) [74]. Conversely, individuals who are not eligible may still receive the benefits associated with these programs. Based on our results, it cannot be determined what part of an intersectional difference in benefit receipt is due to some social groups being more often eligible for benefit receipt, having a higher application rate for benefit programs, or having a higher success rate when applying for benefits. A noteworthy recent advance in studies of benefit receipt involves the use of machine learning models to predict eligibility [75]. However, this avenue of research requires further refinement and validation before it can be effectively employed in empirical studies focusing on benefit receipt.

The present study is the first that comprehensively explored intersectional inequalities in benefit receipt quantitatively. Substantial intersectional effects were found in benefit incidence. This means that some groups are disproportionately vulnerable to benefit receipt while others are relatively advantaged, and that prior research on inequalities in benefit receipt has under- and overestimated the incidences for these intersectional groups. Our results have a direct implication for social policy: to understand inequality in benefit receipt, it is important to consider the interrelatedness of various social characteristics individuals might have.

## Supporting information

**S1 File. Technical description of the analytical strategy.**
(PDF)

**S2 File. MAIHDA results.**
(PDF)

**S3 File. Additional analyses.**
(PDF)

## Author Contributions

**Conceptualization:** Jos Slabbekoorn, Ineke Maas, J. Cok Vrooman.

**Data curation:** Jos Slabbekoorn.

**Formal analysis:** Jos Slabbekoorn.

**Funding acquisition:** J. Cok Vrooman.

**Investigation:** Jos Slabbekoorn.

**Methodology:** Jos Slabbekoorn.

**Supervision:** Ineke Maas, J. Cok Vrooman.

**Visualization:** Jos Slabbekoorn.

**Writing – original draft:** Jos Slabbekoorn.

**Writing – review & editing:** Jos Slabbekoorn, Ineke Maas, J. Cok Vrooman.

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
