## [Decision Letter · Decision Letter 0]

2 Jan 2024

PONE-D-23-17041Intersectionality and Benefit Receipt: The Interplay between Education, Gender, Age and Migration BackgroundPLOS ONE

Dear Dr. Slabbekoorn,

Thank you for submitting your manuscript to PLOS ONE. After careful consideration, we feel that it has merit but does not fully meet PLOS ONE’s publication criteria as it currently stands. Therefore, we invite you to submit a revised version of the manuscript that addresses the points raised during the review process. Please read carefully the reviewers comments. Especially regarding your justification for which variables to include/exclude from the analyses. I agree with the reviewers that it is not necessary to produce new analyses, but you need to explain and clarify your methodological and theoretical decisions.

We look forward to receiving your revised manuscript.

Kind regards,

Eyal Bar-Haim

Academic Editor

PLOS ONE

Journal Requirements:

Reviewers' comments:

Reviewer's Responses to Questions

**Comments to the Author**

1. Is the manuscript technically sound, and do the data support the conclusions?

Reviewer #1: Yes

Reviewer #2: Yes

2. Has the statistical analysis been performed appropriately and rigorously? 

Reviewer #1: Yes

Reviewer #2: Yes

3. Have the authors made all data underlying the findings in their manuscript fully available?

Reviewer #1: No

Reviewer #2: No

4. Is the manuscript presented in an intelligible fashion and written in standard English?

Reviewer #1: Yes

Reviewer #2: Yes

5. Review Comments to the Author

Reviewer #1: Review, Manuscript ID PONE-D-23-17041

Report on “Intersectionality and Benefit Receipt: The Interplay between Education, Gender, Age and Migration Background.” This paper examines the benefit receipt using an intersectional approach. It analyses how combinations of demographic variables and education impact advantages or disadvantages in benefit receipt. Using Dutch administrative data, the authors found that intersectional group disparities are more pronounced in social assistance than unemployment insurance. The authors argue that understanding benefit receipt requires considering the intersection of education, gender, age, and migration background factors.

In general, the article holds significant potential for valuable contributions. However, several concerns need to be addressed, particularly regarding the argumentation and explanation. Below, I outline my concerns and provide suggestions to enhance the article's suitability for publication in PLOS ONE.

A. Theory and argumentation

1. As stated above, my main concern relates to the argumentation and is both relevant to unemployment benefits and Social Assistance. When focusing on receiving unemployment benefits, eligibility for such allowances is contingent upon prior employment and income. Therefore, the assumption that immigrants are more likely to receive higher benefits presupposes a similar entitlement to benefits between native-born individuals and immigrants. The problem lies in our inability to differentiate between eligibility, where immigrants may have had fewer entitlements and the actual need for assistance due to unemployment.

In certain countries, immigrants are less likely to qualify for unemployment-related benefits because they fail to meet the threshold requirements. , It is commendable that the study examines two types of benefits, but there is potential for further exploration. Instead of solely focusing on the receipt of benefits, exploring eligibility as a first step may be worthwhile. This could involve considering factors such as previous continuous employment spanning several months, thereby basing the analysis on a sample of eligible individuals.

2. Similarly, could different groups have higher eligibility for social assistance due to, for example, family size? In that regard, a more detailed explanation is required for the benefit eligibility. Is it family or household-based; how does the household size is taking into account? The paper read, "Social assistance benefit provides financial support to adults in households whose combined income is below the statutory social minimum and whose assets (including housing property) do not exceed a certain threshold ."This does not give enough information about entitlement. In addition, I am wondering about changes over time in the eligibility and if there were any changes in the eligibility for immigrants.

3. The paper's literature review on unemployment primarily focuses on the concept of being without a job and often overlooks the discussion of entitlement to unemployment benefits. This aspect is only mentioned concerning age, disregarding a significant portion of literature relevant to other demographics receiving unemployment benefits. This oversight limits the comprehensive understanding of the topic and should be addressed to encompass a broader range of explanations for receiving benefits.

4. The structure of the paper could benefit from a clearer organization. It might have been more effective to provide a detailed explanation of the theory of intersectionality before discussing the specific main effects, such as education, gender, age, and ethnicity. Alternatively, the paper could have included a section after the introduction that outlines how the content will be divided to enhance the overall coherence and flow of the argument.

5. The literature review presents various arguments regarding predicting the effects of key factors such as education, gender, age, and ethnicity. However, it is important to note that some of these arguments lack sufficient substantiation in the existing literature:

- “Thus, it is expected that lower educated people are generally more likely to receive a benefit than higher educated people”.

- "All in all, women face several disadvantages that increase their likelihood of unemployment and the risk of benefit receipt. Therefore, it is expected that women are, in general, more likely to receive a benefit than men".

- "On these grounds, it is expected that people with a migration background are more likely to receive a benefit than people without a migration background. Additionally, it is expected the incidence of benefit receipt to depend on country of origin, and to be highest among first generation migrants, and lower among second generation migrants."

These assumptions are not always well-founded. In many cases, they ignore entitlements, especially when talking about unemployment benefits. It is true, for example, that there is a higher chance of unemployment for certain groups, but this does not mean that they are more entitled to unemployment benefits.

6. Much of the theoretical explanation focuses on labor market integration rather than eligibility or the utilization of allowances. This is particularly evident when comparing immigrants and natives. Therefore, it is valuable to explore further studies in the field of social work that specifically concentrate on these distinct groups. Such an approach would provide a more comprehensive understanding of the literature review.

B. Methods, Data and Sample

1. I am not an expert on the MAIHDA research method for examining intersectionality, but I wonder if this method may overlook specific contextual factors relevant to the case under investigation. The study seems to lack control over variables such as marital status or the number of children within the sample. Since eligibility for benefits is contingent upon these family contexts, the absence of such controls may lead to unequal comparisons. For instance, immigrant families are often larger, and eligibility for benefits could be influenced by family size. Similarly, if women are less likely to work due to accompanying their partners, they would have reduced eligibility for unemployment benefits. Considering these differences in the characteristics of different groups, I am curious about how the models account for and address these variations.

2. Immigrants from Eastern and Central Europe are categorized together. It would be beneficial to explore how countries that have joined the EU in recent years are classified. This is due to the potential association between EU membership and eligibility for benefits.

3. There needs to be clarity regarding including individuals over 60 in the sample. The sample description initially states that only individuals under 60 are included (page 15), but the results include people up to 69. It's possible that it was meant that people under 60 were included if they were under the age of 60 in the first year, but clarity on this matter is required.

C. Results

1. Based on the results presented in Evens et al.'s (2018) article, creating a graph that provides more detailed information about the intersectional groups would be beneficial. The current findings are solely described in tables, limiting our understanding of the specific groups involved. It would be interesting to visualize these groups divided by gender, ethnic group, and education, showing additional advantages or disadvantages beyond what the additive model predicts. This graph would offer a clearer picture of the dynamics at play and what is presented in Figure 3.

2. Table 2 does not include young people (25-34), and why they are omitted from the table is unclear.

D. Conclusion and discussion

1. In the conclusion section, the article emphasizes the application of this method to assess its relevance to the context rather than delving into comprehensive explanations for the obtained results. While the data indicates the presence of strata that experience either adversity or advantages through intersectionality, there is a lack of clarity regarding the specific identities of these groups and the underlying reasons for their circumstances. As a result, the article leaves readers with limited knowledge about the individuals affected and the factors contributing to their situations.

2. Furthermore, the section discussing the study's limitations in the conclusions appears to be quite formulaic. While it is crucial to acknowledge and address limitations, the specific limitations highlighted, such as the levels of measurement and the challenges in measuring variables, seem somewhat arbitrary. In the context of immigration, it would be more pertinent to explore factors like the duration since immigration or the age at which individuals immigrated. Additionally, it is essential to clarify that the article primarily aims to present cases of intersectionality rather than delve into the underlying causes of such cases.

Overall, the paper is very interesting and has an important promotional contribution to the literature. I believe that the paper could have done more in explaining the underlying cases that are more common to enjoy or suffer from intersectional and not limiting its contribution to the exploratory study of intersectional inequalities in benefit receipt.

Additional references that could be included in the paper:

Renema, J. A., & Lubbers, M. (2019). Welfare-based income among immigrants in the Netherlands: Differences in social and human capital. Journal of Immigrant & Refugee Studies, 17(2), 128-151.

Yu, Y. C. (2023). Precarious welfare‐to‐work transitions in a segmented labour market: Evidence from the Netherlands. International Journal of Social Welfare.

Strockmeijer, A., de Beer, P., & Dagevos, J. (2020). Explaining differences in unemployment benefit takeup between labour migrants and Dutch native workers. International Social Security Review, 73(2), 75-99.

Reviewer #2: I have enjoyed reading the paper "Intersectionality and Benefit Receipt: The Interplay between Education, Gender, Age and Migration Background" and found the analysis carefully conducted and well drafted up. I feel that the paper is probably close to being publishable in its current form, though I would like to draw the authors' attention to the following points, which may still need to be addressed:

1) Construction of the dependent variable: I am having some doubts about how the authors constructed the two benefit receipt variables given their sample selection criteria. From what I understand, the two benefit receipt variables, which are the dependent variables in their model, measure whether an individual received benefits at least once for three consecutive months at any time during the observation period (Subsection 2.2.1). The observation period varies across individuals: it can reach 14 years (2006-2019), but is shorter for individuals who either age into or out of the sample (e.g. turn 25 or 60 during the observation period) or who enter or leave the sample for other reasons (e.g. presumably migration, death). If this is correct, benefit receipt probabilities will be biased downwards for groups with systematically shorter observation periods (young people, old people, migrants, etc) because it is less likely – ceteris paribus – that these people will be observed as receiving benefits during the observation period. For example, the cohort of young people who turned 25 in late 2016 (and are therefore just observed for the required minimum of three years) should by construction have lower receipt probabilities than the cohorts who turned 25 in earlier years and are therefore observed for longer. It seems that a simple solution to this problem would be to construct the sample (and the dependent variables) from a balanced panel, e.g. to restrict the sample to people who have been observed for, say, at least 5 years (60 months) and then restrict the observation period to the first 60 months that are observed.

2) Motivation: The authors motivate the paper by stating that “Prior research on benefit receipt has typically relied on the implicit assumption that the advantages and disadvantages from various social dimensions are independent of one another” (Introduction, para #2). I don’t think this assessment is fair. For example, in papers on state dependence in social assistance benefit receipt, some of which the authors cite in their paper, it is common practice to estimate separate models for migrants and natives (notably in the papers on Sweden) and for men and women. Many of these models also include interaction terms between different individual characteristics. The reason for why such specifications have been chosen is precisely to permit for differences in the association between benefit receipt and education or age by gender and migrant status. The same comment applies to the statement in the Conclusion that earlier studies “assume that disadvantage or advantage accumulates in an additive manner” and have therefore over-/underestimated benefit receipt rates.

3) Added value of the model: it seems like the easiest and most intuitive way of exploring the intersectionality of different personal characteristics and their relationship to benefit receipt would to be simply calculate and compare benefit receipt rates for different subgroups from the raw data. Put differently: it is not entirely obvious what the value is of predicting the incidence of benefit receipt, as done in Figure 2; these receipt rates can be easily just calculated from the data to illustrate the importance of intersectionality. It seems to me that the added value of the model over such simple cross-tabulations is that it allows exploring in a more systematic way the nature and types of intersectionality. But it would be useful if authors could be more explicit on what additional insights the model brings over and above calculating simple subgroup-specific receipt rates, and maybe even how the model estimates compare to those empirical rates (maybe they could be included in Figure 2 for comparison)? It appears that to reach the conclusion given at the very end of the paper, namely that “to understand inequality in benefit receipt, it is important to consider the interrelatedness of various social characteristics individuals might have”, a simple descriptive comparison of empirical receipt rates across subgroups defined based on the same variables would have been sufficient.

4) Other drivers / correlates of benefit receipt: besides the four dimensions that the authors use to define their strata (sex, age, migrant background, education), a number of other personal and household characteristics are known to be closely associated with the probability of receiving income support benefits, including household type and the number of kids (notably for social assistance), location of residence, and health status. At least the two former variables can probably be identified in the data, and they are often included when modelling income support benefits. It would be useful if the authors could expand on whether it would be possible to account for these variables in the analysis. I imagine that including additional strata would greatly complexify the analysis (though maybe the number of migration-related outcomes could be reduced). But how does the analysis in its current form account for these variables? Or else, does the variable for age or migrant status partly pick up differences in household type (single / couple, with or without kids) for people of different ages / between migrants and natives? I think this aspect merits discussion.

Some additional minor comments:

1) Page 3, para #2: the authors point to the scientific relevance of their results in contributing to a better understanding of social inequalities. Maybe it would be worth including also a sentence on the potential relevance for policy making: to the extent that intersectionality matters, the effectiveness of support policies can be increased by targeting them not just at broader population groups (e.g. young people or migrants) but much more specific subgroups (e.g. young people who are also migrants, have low educational attainment or reduced work capacity).

2) Throughout the paper, the authors seem to use “unemployed” as a synonym for being out of work / jobless, which is not correct. Unemployed people are only those who are available and actively looking for work. In that sense, the reference in Section 1.1.2. to women who are “voluntarily unemployed and live in a traditional household where the man is the sole breadwinner” is not correct. Most of these women are probably economically inactive, not unemployed, because they likely do not look for work. Similarly, in the same paragraph, it is not clear that women face a larger risk of unemployment. They are more likely to be out of work / jobless. I would suggest revising the terminology here and possibly in other parts of the paper.

3) Page 7, bottom paragraph: Young people are a priority group for ALMPs in all EU countries (not just the Netherlands) under the EU Youth Guarantee scheme, which is implemented by the member states.

4) Page 10, note to Figure 1: “disadvantaged social characteristics” should probably rather be “disadvantageous social characteristics”

5) Page 14, Section 2.2.1: I think it would be useful to point out that the benefit receipt variable, as constructed in the paper, ignores any information on the duration (or frequency) of benefit receipt over the observation period. In other words, it looks at the incidence of benefit receipt, not at its intensity. In this sense, the analysis differs from those in other studies of unemployment benefit / social assistance dynamics, in which the duration of benefit receipt, and possible repeated spells, are key outcome of interest.

6) Page 14, Section 2.2.1: unemployment benefits are individual-level benefits, but social assistance is usually household level. Maybe the authors could clarify whether / how for social assistance the individual-level benefit variable was constructed from household-level information on benefit receipt.

7) P. 18, last paragraph: typo – “calculated” should be “calculate”

8) P.22, note to Figure 2: I don’t find the figure fully self-explanatory. “Rank” presumably gives the rank of a specific stratum when strata are sorted by the predicted benefit receipt rate in ascending order? Maybe worth clarifying.

9) I believe Figure 3 was missing from the document shared with the referees for review.

10) Page 23. In the discussion of the average marginal effects, maybe it would be useful to compare the size of these effects to the average benefit receipt rate, or the simple differences in receipt rates between broader groups, to give readers a sense of the magnitude.

11) Page 26, bottom paragraph: the term “long-term unemployment insurance benefits” is not clear.

6. PLOS authors have the option to publish the peer review history of their article (what does this mean?). If published, this will include your full peer review and any attached files.

Reviewer #1: No

Reviewer #2: No

---

## [Author Response · Author response to Decision Letter 0]

1 Mar 2024

Dear Editor and Reviewers,

We would like to express our sincere gratitude to the editor and reviewers for their valuable and insightful comments. We have carefully considered all the feedback and have made a sincere effort to address each point.

In the following, we outline the specific revisions we have made in response to the feedback provided. We believe these changes significantly enhance the clarity and quality of the manuscript. Based on the revisions made in response to the feedback, we hope the manuscript is now ready for publication in PLOS ONE.

Reviewer #1: Review, Manuscript ID PONE-D-23-17041

Report on “Intersectionality and Benefit Receipt: The Interplay between Education, Gender, Age and Migration Background.” This paper examines the benefit receipt using an intersectional approach. It analyses how combinations of demographic variables and education impact advantages or disadvantages in benefit receipt. Using Dutch administrative data, the authors found that intersectional group disparities are more pronounced in social assistance than unemployment insurance. The authors argue that understanding benefit receipt requires considering the intersection of education, gender, age, and migration background factors.

In general, the article holds significant potential for valuable contributions. However, several concerns need to be addressed, particularly regarding the argumentation and explanation. Below, I outline my concerns and provide suggestions to enhance the article's suitability for publication in PLOS ONE.

A. Theory and argumentation

1. As stated above, my main concern relates to the argumentation and is both relevant to unemployment benefits and Social Assistance. When focusing on receiving unemployment benefits, eligibility for such allowances is contingent upon prior employment and income. Therefore, the assumption that immigrants are more likely to receive higher benefits presupposes a similar entitlement to benefits between native-born individuals and immigrants. The problem lies in our inability to differentiate between eligibility, where immigrants may have had fewer entitlements and the actual need for assistance due to unemployment.

In certain countries, immigrants are less likely to qualify for unemployment-related benefits because they fail to meet the threshold requirements. It is commendable that the study examines two types of benefits, but there is potential for further exploration. Instead of solely focusing on the receipt of benefits, exploring eligibility as a first step may be worthwhile. This could involve considering factors such as previous continuous employment spanning several months, thereby basing the analysis on a sample of eligible individuals.

We thank the reviewer for this valuable suggestion, and agree that entitlements were previously not sufficiently addressed in our paper. In the revised manuscript we have incorporated arguments about entitlements and eligibility in our theoretical argumentation. Additionally, we have extended the description about entitlements in the introduction of the paper. 

Although the suggestion to incorporate entitlements in the analysis is interesting, we think that such an analysis goes beyond the aim of this paper to assess intersectional inequalities in benefit receipt. Moreover, appropriately modelling eligibility would require the estimation of stochastic predictive models, using indicators for the entry requirements for these benefits. For unemployment insurance this is particularly troublesome because some of the key indicators (e.g. non-voluntary termination of employment) are not readily available. Therefore, we deem it impossible to accurately approximate eligibility in the present study. To address this point, we have included an extensive discussion of this limitation in the conclusion of our manuscript:

“Furthermore, it is imperative to contextualize the results presented in this paper in light of one notable qualification. Due to data limitations, this study cannot take into account individuals’ eligibility for benefit programs. Our assumption now is that people who do not receive benefits do not have a need for it because they have a paid job or other resources to get by. However, some individuals meeting the criteria for either social assistance or unemployment insurance might not have received a benefit for various reasons (i.e., not applying, not obtaining, and not qualifying) [74]. Conversely, individuals who are not eligible may still receive the benefits associated with these programs. Based on our results, it cannot be determined what part of an intersectional difference in benefit receipt is due to some social groups being more often eligible for benefit receipt, having a higher application rate for benefit programs, or having a higher success rate when applying for benefits. A noteworthy recent advance in studies of benefit receipt involves the use of machine learning models to predict eligibility [75]. However, this avenue of research requires further refinement and validation before it can be effectively employed in empirical studies focusing on benefit receipt.”

2. Similarly, could different groups have higher eligibility for social assistance due to, for example, family size? In that regard, a more detailed explanation is required for the benefit eligibility. Is it family or household-based; how does the household size is taking into account? The paper read, "Social assistance benefit provides financial support to adults in households whose combined income is below the statutory social minimum and whose assets (including housing property) do not exceed a certain threshold. "This does not give enough information about entitlement. In addition, I am wondering about changes over time in the eligibility and if there were any changes in the eligibility for immigrants.

The entry requirements in the Netherlands are household based and are to a limited extent affected by household type. For instance, single parents and cohabitating partners are currently allowed to have more savings/wealth (EUR 15.150) than one-person households (EUR 7575). The number of children (under 18 years of age) does not affect the eligibility requirements. In the Netherlands additional child support benefits are available which do not affect the eligibility for social assistance and unemployment insurance. 

During our observation period, some changes to eligibility requirements for social assistance and unemployment insurance were enacted. Notably, the 2015 Participation Act introduced stricter social assistance requirements for all Dutch residents. Additionally, the 2013 citizenship act mandated Dutch language proficiency for social assistance applicants. However, these legislative changes likely have a negligible impact on our results for two reasons: 1) The Participation Act was enacted within our observation period, and our analysis considers benefit receipt incidence across the entire period. 2) The citizenship act primarily affects asylum-seeking migrants, who represent a very small portion of our sample.

The restriction of social assistance and unemployment insurance to registered residents of the Netherlands does not pose a limitation since our analysis relies on register data, which only include this group. Additionally, while EEA (European Economic Area) citizens' unemployment insurance eligibility considers their EEA employment history, we did not observe a higher incidence of unemployment insurance benefit receipt among individuals with a European origin. We have clarified this in the second paragraph on page 5. 

3. The paper's literature review on unemployment primarily focuses on the concept of being without a job and often overlooks the discussion of entitlement to unemployment benefits. This aspect is only mentioned concerning age, disregarding a significant portion of literature relevant to other demographics receiving unemployment benefits. This oversight limits the comprehensive understanding of the topic and should be addressed to encompass a broader range of explanations for receiving benefits.

As also mentioned in our response to the first point of feedback, we have included arguments about entitlement throughout our theoretical argumentation. For example, we incorporated entitlements in the discussion of migration background differences on page 8: 

“There are considerable differences between migrant groups and natives, which may translate into diverging incidence rates of benefit receipt [46,47]. On the one hand, poorer language proficiency of some groups of first generation migrants may on average lead to a lower take- up of benefits. However, findings regarding the take-up of benefits are mixed; where some studies find comparable take-up of benefits between migrants and natives [48,49], others find lower take-up of benefits [50]. Recent migrants may also have a too short employment history to be eligible for unemployment benefits, but with the exception of those who are in the Netherlands shorter than 3 months, all legal migrants are eligible for social assistance. We do not expect that eligibility and differential take-up will lead to a substantial lower incidence of benefit receipt for migrants compared to natives.”

4. The structure of the paper could benefit from a clearer organization. It might have been more effective to provide a detailed explanation of the theory of intersectionality before discussing the specific main effects, such as education, gender, age, and ethnicity. Alternatively, the paper could have included a section after the introduction that outlines how the content will be divided to enhance the overall coherence and flow of the argument.

We chose the current structure of the theory section to show how intersectional argumentation can be integrated in the existing literature on social inequalities in benefit receipt. This structure comes with the additional advantage that we first establish which social groups are disadvantaged with regard to benefit recipiency, which then aids the intersectional argumentation. We have incorporated the reviewer’s suggestion to include a section in which we outline the structure of our theoretical argumentation.

5. The literature review presents various arguments regarding predicting the effects of key factors such as education, gender, age, and ethnicity. However, it is important to note that some of these arguments lack sufficient substantiation in the existing literature:

- “Thus, it is expected that lower educated people are generally more likely to receive a benefit than higher educated people”.

We clarified in the text that the expectation is based on the preceding reasoning. In this case, we don’t think that there are differences in entitlements between higher and lower educated that would work against our explanation (higher educated may be richer and therefore not entitled to social assistance when unemployed for a longer period – but this only strengthens our expectation).

- "All in all, women face several disadvantages that increase their likelihood of unemployment and the risk of benefit receipt. Therefore, it is expected that women are, in general, more likely to receive a benefit than men".

As suggested by the reviewer (below), women who are temporarily out of the labor market (when they decide to become a stay-at-home mom) are not entitled to unemployment benefits. They usually will also not apply for these benefits (or for social assistance) as long as they have a working partner. But in case of divorce these women are very likely to receive social assistance. Because of their limited time in the labor market, they will not be eligible for unemployment benefits, but they are not likely to immediately find employment, especially if they have young children. We explained this more clearly in the new text, at page 7.

- "On these grounds, it is expected that people with a migration background are more likely to receive a benefit than people without a migration background. Additionally, it is expected the incidence of benefit receipt to depend on country of origin, and to be highest among first generation migrants, and lower among second generation migrants."

For migrants we also added additional information on eligibility and possible increased non-take up. However, we don’t think that these outweigh the disadvantages that migrants face on the labor market.

These assumptions are not always well-founded. In many cases, they ignore entitlements, especially when talking about unemployment benefits. It is true, for example, that there is a higher chance of unemployment for certain groups, but this does not mean that they are more entitled to unemployment benefits.

As specified above, we have included more argumentation for the abovementioned cases.

6. Much of the theoretical explanation focuses on labor market integration rather than eligibility or the utilization of allowances. This is particularly evident when comparing immigrants and natives. Therefore, it is valuable to explore further studies in the field of social work that specifically concentrate on these distinct groups. Such an approach would provide a more comprehensive understanding of the literature review.

As mentioned in our response to prior points, we have extended our theoretical argument about entitlements and eligibility throughout the theory section. We incorporated references from the field of social work, including several of those suggested by the reviewer.

B. Methods, Data and Sample

1. I am not an expert on the MAIHDA research method for examining intersectionality, but I wonder if this method may overlook specific contextual factors relevant to the case under investigation. The study seems to lack control over variables such as marital status or the number of children within the sample. Since eligibility for benefits is contingent upon these family contexts, the absence of such controls may lead to unequal comparisons. For instance, immigrant families are often larger, and eligibility for benefits could be influenced by family size. Similarly, if women are less likely to work due to accompanying their partners, they would have reduced eligibility for unemployment benefits. Considering these differences in the characteristics of different groups, I am curious about how the models account for and address these variations.

The reviewer is right that marital status is a potential explanation for benefit receipt differences between intersectional groups. Besides that, married women are more often not working, there are also more relaxed entry requirements for social assistance for cohabitating and married couples. It is less likely that the number of children explains intersectional differences. Although there are considerable differences in the number of children between intersectional strata, the number of children does not make people more susceptible or eligible for benefit receipt. 

However, the main aim of our study is to describe, not to explain, inequalities between intersectional groups (which are referred to as intersectional strata in the context of this methodology). MAIHDA models are a good instrument to do so and are usually applied in this descriptive way (compare for example 1, 2, 3 at the end of this letter). This is not to say that MAIHDA models could not be extended with spurious, mediating and moderating effects. We think that such expansions would be a valuable contribution to the literature on social benefit receipt, but they do not fit the scope of this paper. It would require additional theoretical reasoning and a substantial expansion of the method section. We added a paragraph in the conclusion explaining that adding more variables is possible, but we don’t do so, because we first want to establish to what extent intersectional effects exist.

2. Immigrants from Eastern and Central Europe are categorized together. It would be beneficial to explore how countries that have joined the EU in recent years are classified. This is due to the potential association between EU membership and eligibility for benefits.

The eligibility rules for unemployment benefits and social assistance are generally similar for migrants from within and out

---

## [Decision Letter · Decision Letter 1]

17 Sep 2024

Intersectionality and Benefit Receipt: The Interplay between Education, Gender, Age and Migration Background

PONE-D-23-17041R1

Dear Dr. Slabbekoorn,

We’re pleased to inform you that your manuscript has been judged scientifically suitable for publication and will be formally accepted for publication once it meets all outstanding technical requirements.

Kind regards,

Eyal Bar-Haim

Academic Editor

PLOS ONE

Additional Editor Comments (optional):

Reviewers' comments:

Reviewer's Responses to Questions

**Comments to the Author**

1. If the authors have adequately addressed your comments raised in a previous round of review and you feel that this manuscript is now acceptable for publication, you may indicate that here to bypass the “Comments to the Author” section, enter your conflict of interest statement in the “Confidential to Editor” section, and submit your "Accept" recommendation.

Reviewer #1: All comments have been addressed

Reviewer #3: All comments have been addressed

2. Is the manuscript technically sound, and do the data support the conclusions?

Reviewer #1: Yes

Reviewer #3: Yes

3. Has the statistical analysis been performed appropriately and rigorously? 

Reviewer #1: Yes

Reviewer #3: Yes

4. Have the authors made all data underlying the findings in their manuscript fully available?

Reviewer #1: Yes

Reviewer #3: Yes

5. Is the manuscript presented in an intelligible fashion and written in standard English?

Reviewer #1: Yes

Reviewer #3: Yes

6. Review Comments to the Author

Reviewer #1: The authors have diligently addressed all the comments and suggestions provided by both myself and the other reviewer. I am confident that the paper has undergone significant improvements and is now well-prepared for publication. Furthermore, I believe that the paper will be of great interest to the readership of the journal.

Reviewer #3: The authors have provided comprehensive and thorough responses to the reviews they received, addressing all points raised in considerable detail. Their explanation regarding the limitations of addressing benefit eligibility is both reasonable and understandable, given the scope of their research and the constraints involved. They have demonstrated a clear understanding of the issues presented and have carefully adjusted their work to align with the reviewers' feedback where appropriate. After reviewing their responses and revisions, it is my impression that the paper now meets the necessary standards and is suitable for publication in PLOS ONE. The quality of their work is commendable.

7. PLOS authors have the option to publish the peer review history of their article (what does this mean?). If published, this will include your full peer review and any attached files.

Reviewer #1: **Yes: **Debora Pricila Birgier

Reviewer #3: No

---

## [Editor Report · Acceptance letter]

5 Nov 2024

PONE-D-23-17041R1 

PLOS ONE

Dear Dr. Slabbekoorn, 

I'm pleased to inform you that your manuscript has been deemed suitable for publication in PLOS ONE. Congratulations! Your manuscript is now being handed over to our production team.

Kind regards, 

on behalf of

Dr. Eyal Bar-Haim 

Academic Editor

PLOS ONE